# VRNN's got a GAN: Generating Time Series using Variational Recurrent Neural Models with Adversarial Training

## Abstract

Time-series data generation is a machine learning task growing in popularity, and has been a focus of deep generative methods. The task is especially important in fields where large amounts of training data are not available, and in applications where privacy preservation using synthetic data is preferred. In the past, generative adversarial models (GANs) were combined with recurrent neural networks (RNNs) to produce realistic time-series data. Moreover, RNNs with time-step variational autoencoders were shown to have the ability to produce diverse temporal realizations. In this paper, we propose a novel data generating model, dubbed VRNN-GAN, that employs an adversarial framework with an RNN-based Variational Autoencoder (VAE) serving as the generator and a bidirectional RNN serving as the discriminator. The recurrent VAE captures temporal dynamics into a learned time-varying latent space while the adversarial training encourages the generation of realistic time-series data. We compared the performance of VRNN-GAN to state-of-the-art deep generative methods on the task of generating synthetic time-series data. We show that VRNN-GAN achieves the best predictive score across all methods and yields competitive results in other well-established performance measures compared to the state-of-the-art.

## 1 Introduction

Time-series data is found in a large number of diverse fields and applications. Broadly described as recordings of time-indexed values, such data often appears as continuous or categorical sequences, and may contain patterns or behaviours that vary over time Montanez et al. (2015). Some examples of time-series data include stock prices over a period of time, a person's heart-rate throughout their workout session, and the amount of hits a website gets over a specific time frame. Time-series analysis has important applications in finance Cochrane (2005), biological sciences Peng et al. (1995), and marketing science Dekimpe & Hanssens (2000). Important world events such as the COVID-19 are also often recorded as time-series data (e.g., case counts), in which modelling work can be applied to forecast disease severity and prevention Ayoobi et al. (2021). Machine learning, and specifically, deep neural networks have been popular and successful in modelling and analyzing time-series data due to their expressiveness and use of non-linear mechanisms Zhang & Qi (2005); Abiodun et al. (2018).

When data collection becomes time consuming, expensive, or regulated, data availability can become a bottleneck for deep learning techniques Roh et al. (2019). For example, in clinical research, the large focus on patient privacy and safety has resulted in smaller available datasets, which has slowed the development of advanced modelling initiatives Goncalves et al. (2020). In addition, missing values arising, e.g., from recording errors over time Pratama et al. (2016), are a common problem in time-series data. Therefore, due to the often limited amount of data, generating realistic time-series is essential.

Variational Autoencoders (VAEs) and Generative Adversarial Networks (GANs) are deep generative models that have been combined as hybrids and applied with recurrent neural networks (RNNs) to generate sequence data Zhao et al. (2018); Jin et al. (2018) and molecule representations Jin et al. (2018). In particular,

VAE-GAN hybrid methods generally focus on using a VAE to learn a latent space suitable for generating samples, and a Discriminator that regularizes the posterior and prior distributions of the latent sequence representations. Results from these hybrid methods have shown improved latent spaces and diversity of results Zhao et al. (2018)

Motivated by the success of approaches that combine VAEs, RNNs, and GANs for sequential applications, we propose the *VRNN-GAN*, a novel solution for generating time-series data by incorporating the stochastic variational RNN framework from Chung et al. (2015) with an adversarial approach. Specifically, VRNN-GAN is similar to past VAE-GAN hybrids in that it uses a VAE decoder to generate data and a discriminator network and GAN objective for adversarial training. In contrast to previous methods, VRNN-GAN samples from the recurrent time-dependent latent space learned from the VRNN to generate time-series data, and uses adversarial training to encourage the generated samples to be similar to the real time-series data.

Our experiments demonstrate the advantages of the VRNN-GAN using both synthetic and real-world data to train our model. Specifically, we show that VRNN-GAN outperforms state-of-the-art methods for time-series data generation such as Yoon et al. (2019) and Mogren (2016), especially in terms of predictive scoring. We are particularly encouraged by the latter since predictive scoring measures how successful a time-series model trained on generated data can generalize to real data, which indicates that the generated and real datasets have similar distributions.

## 2 Related Work

As deep generative models continue to gain traction in various data-domains, time-series and sequential data generation have emerged as an important area of development in deep learning Brophy et al. (2021). In this part, we provide a review of the relevant VAE-based approaches for sequential data as well as related time-series generating methods.

### 2.1 Variational adversarial models for sequences

Jin et al. (2018) proposed a graph-to-graph translation model for generating graph representations of molecules. Specifically, they use RNNs to model and generate the tree structures of the molecules, where as a VAE is used to learn the latent space for generating samples. Adversarial training is incorporated by applying the discriminator to continuous representations of the decoded samples. Zhao et al. (2018) propose the Adversarially Regularized Autoencoders (ARAE) where they apply the Wasserstein autoencoder (WAE) framework using the GAN objective as the regularizer between the posterior and prior distributions for the latent space to model discrete sequences (with RNNs incorporated in the encoder and decoder architectures). Gu et al. (2018) propose DialogWAE, which extends the WAE framework with an adversarial regularizer by incorporating a conditional context element for generating text responses based on dialogue context.

Akbari & Liang (2018) propose a semi-recurrent VAE-GAN architecture that utilizes convolutional neural networks (CNN) in the generator and discriminator models. Similarly to our approach, they use a time-step based VAE objective to learn the time-dependent latent space. However, unlike the VRNN-GAN, this method does not incorporate an RNN in its variational component.

VRNN-GAN is motivated by these frameworks, yet focuses on using the flexible, recurrent time-dependent VAE mechanism to sample latent variables and generate time-series data, rather than relying on a single latent variable from a non time-dependent or non recurrent VAE to represent a whole sequence. Moreover, VRNN-GAN's adversarial training uses the discriminator network to classify the decoded time-series data generated from the prior and posterior distributions rather than the latent variables themselves.

### 2.2 Time-series generating models

In previous work, VAE models have been used for time-series generation. The Variational Recurrent Neural Network (VRNN) from Chung et al. (2014) was used to generate structured sequential data, including audio signals and hand-writing strokes, and Alias Parth Goyal et al. (2017) extend the VRNN framework by incorporating information from hidden states in a backwards RNN, allowing the latent space to better reflect

future time-series values. The time-dependent VAE mechanism was found to accurately retain the writing styles through the generation sequence in the hand-writing task, and the posterior and conditional prior distributions were found to accurately reflect temporal patterns in audio waveforms. Fabius & Van Amersfoort (2014) proposed the Variational Recurrent Auto Encoder (VRAE), which consists of RNN encoder and decoder networks, where the final hidden state in the RNN encoder is used as input to learn a latent distribution from which the RNN decoder generates data. Desai et al. (2021) proposed TimeVAE, which follows a similiar architecture from VRAE, but using 1-D convolutional neural networks instead of RNNs. However, similarly to the VAE-GAN hybrids mentioned above, VRAE and TimeVAE also lack a time-dependent VAE, and rely on a single latent variable to generate a sequence. Furthermore, VRNN-GAN allows for the inclusion and replacement of the VRNN component with similar stochastic RNNs, e.g., Z-Forcing.

Generative adversarial networks (GANs) were employed for the generation of both discrete and continuous synthetic time-series data Shorten & Khoshgoftaar (2019). While the exact architecture of these models may vary, the $G$ function is typically used to generate a temporal representation of the data (in the observed or latent state), while the $D$ function predicts whether or not an input sequence is generated or real. The Continuous Recurrent Neural Network GAN (C-RNN-GAN) use RNNs in both the generator and discriminator functions to model temporal data Mogren (2016). Yoon et al. (2019) argue that existing GAN methods do not adequately capture the temporal dynamics and correlations found in time-series data, and propose additional training objectives such as next-step prediction and adversarial training on latent codes to improve the learning of time-series data. Esteban et al. (2017) propose RCGAN, a conditional GAN for generating medical time-series data conditioned on time-static features. RCGAN generates time-series data based on conditional inputs and additionally uses a discriminator that minimizes the negative cross entropy loss per time-step between fake or real outputs, as opposed to discriminating an entire sequence. Yet, this method requires a set of auxiliary information to condition upon.

Unlike the VRNN-GAN, C-RNN-GAN and TimeGAN lack a VAE component, and as a result, do not have a disentangled latent space to sample from. Moreover, VRNN-GAN is not a conditional model, unlike the RCGAN, and does not require additional contextual information to generate data.

## 3  The VRNN-GAN

The VRNN-GAN solution is presented in Figure 1. Specifically, it has two main components: the *generator*, which is based on a combination of VAE and RNN (namely, the VRNN), and the *discriminator*. The VRNN generates time-series samples through its inference and generational models, which are passed to the discriminator along with the real time-series data. The discriminator then attempts to classify the real and generated data accurately, and the errors are then back-propagated to the VRNN, which adjusts its parameters to better generate realistic time-series data. We discuss the two components and detail their architectures. Then, we provide the VRNN-GAN loss function, which we use for training the model.

### 3.1  Recurrent Variational Component

The VRNN component is the same model implemented from Chung et al. (2015). The VRNN uses latent variables in the calculation of the recurring hidden state to model time-series data, learned through a time-wise variational inference approach. This approach was shown to have outperformed RNN models with Gaussian and Gaussian Mixture Model outputs, showing that the inclusion of latent-random variables in the hidden state of RNNs allow for a more expressive generative model.

Let $x_1, x_2, ..., x_n$ be a time-series with $n$ data points and let $z_1, ..., z_n$ be a sequence of the corresponding latent variables. We provide the generation and inference models of the VRNN, which is an adjustment of the VAE model to time-series observations. Figure 2a presents a schematic overview of the generator part whereas Figure 2b shows the inferential model. In the generator, the joint probability factorization is assumed to be of the form:

$$p(x_{\leq T}, z_{\leq T}) = \prod_{t=1}^{T} p(x_t | x_{<t}, z_{\leq t}) p(z_t | x_{<t}, z_{<t}). \tag{1}$$

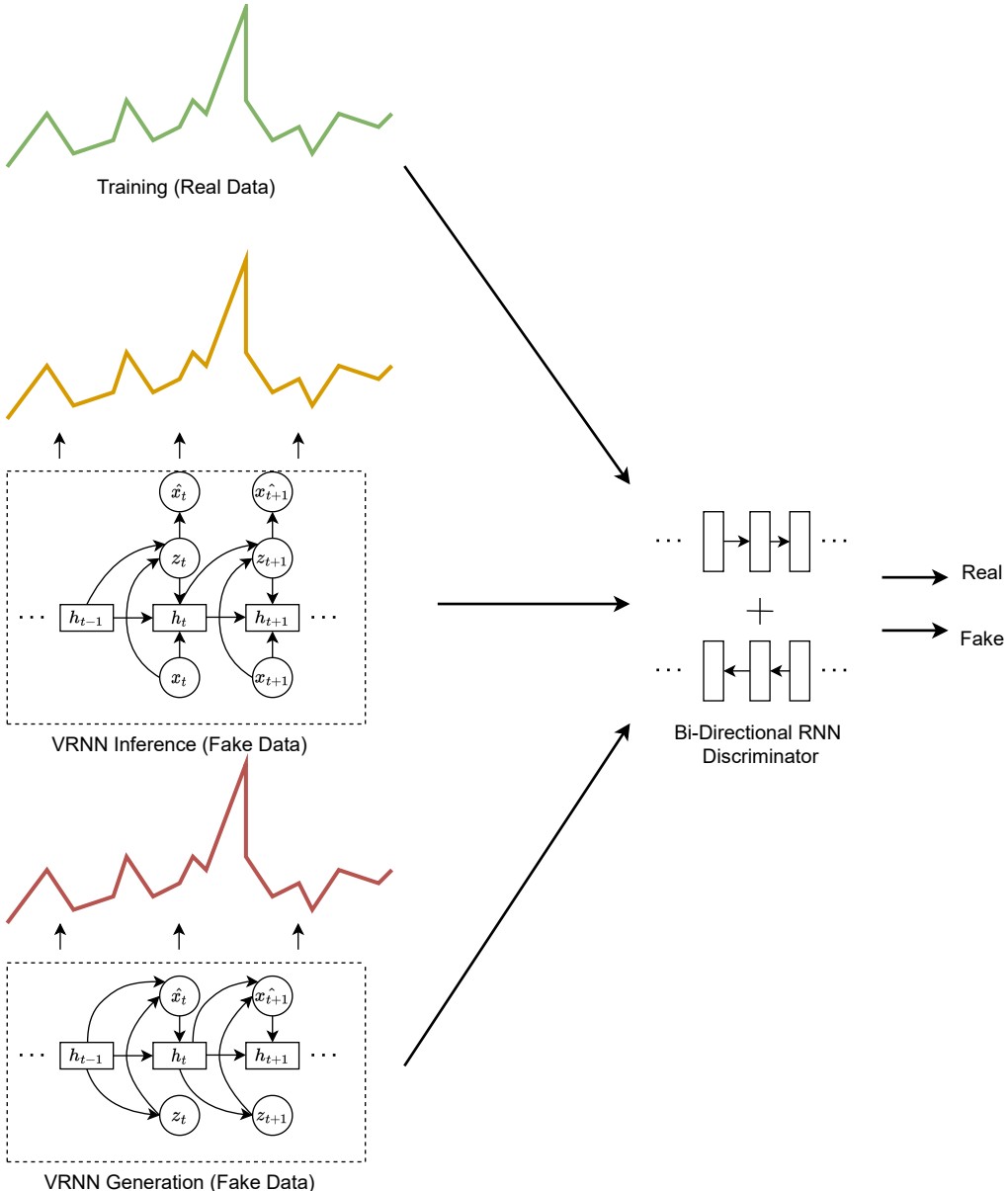

Figure 1: VRNN-GAN Generation and Discrimination Process

The relationships in the generator model between the variables are described as follows:

$$z_t \sim \mathcal{N}(\mu_{0,t}, diag(\sigma_{0,t}^2)), \tag{2}$$

$$[\mu_{0,t}, \sigma_{0,t}] = \varphi_\tau^{prior}(h_{t-1}), \tag{3}$$

$$x_t | z_t \sim \mathcal{N}(\mu_{x,t}, diag(\sigma_{x,t}^2)), \tag{4}$$

$$[\mu_{x,t}, \sigma_{x,t}] = \varphi_\theta^{dec}(\varphi_\phi^z(z), h_{t-1}), \tag{5}$$

where $\varphi_\tau^{prior}, \varphi_\theta^{dec}, \varphi_\phi^z$ are flexible functions. Note that $h_{t-1}$ refers to the hidden state calculated by the RNN model. Variational inference is used to learn a time-varying approximate posterior distribution for the latent variables $z_1, .., z_T$. We let $q_\theta$ denote this approximate posterior distribution. The joint conditional

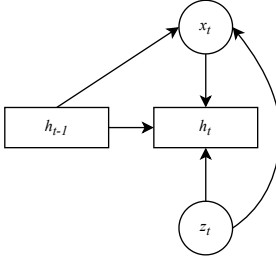

(a) Interactions between the hidden states $h_{t-1}, h_t$, latent variable $z_t$ and output $x_t$ in the generation model

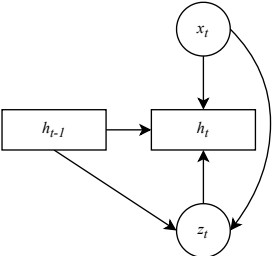

(b) Interactions between the hidden states $h_{t-1}, h_t$, latent variable $z_t$ and output $x_t$ in the inference model

Figure 2: Generation and inference models of the VRNN.

probability factorization is then given by,

$$q_\theta(z_{\leq T}|x_{\leq T}) = \prod_{t=1}^{T} q_\theta(z_t|x_{\leq t}, z_{<t}), \tag{6}$$

$$z_t|x_t \sim \mathcal{N}(\mu_{z,t}, diag((\sigma_{z,t})^2)), \tag{7}$$

$$[\mu_{z,t}, \sigma_{z,t}] = \varphi_\theta^{enc}(\varphi_\phi^x(x_t), h_{t-1}), \tag{8}$$

where $\varphi_\theta^{enc}, \varphi_\phi^x$ are, as before, flexible functions. Note that unlike the latent parameter estimation process in the generation model, the latent parameter estimates in the inference model depend on both $x_t$ and the previous hidden state $h_{t-1}$. The parameters of the distributions of the latent and observed variables are estimated by a per-time unit VAE that operates on the outputs of an RNN cell. In our approach, the choice of the RNN cell is arbitrary; specifically, we use a GRU cell.

## 3.2 Adversarial Component

GAN models are composed of two parts, a generator function $G$ and a discriminator function $D$, conceptualized as two adversaries. In the model proposed by Goodfellow et al. (2014), $D$ and $G$ are two neural networks, where $D$ is a binary classifier that is trained to distinguish between real data (as taken from the training set) and fake data (generated from $G$). As $D$ learns to perform better classifications, $G$ is trained to maximize the error found in $D$ by producing data that is as similar to the real data as possible. The loss function for the GAN model in Goodfellow et al. (2014) is known as the min-max loss, and has the form:

$$\min \max V(G, D) = \mathbb{E}[log(D(x)] + \mathbb{E}[log(1 - D(z)]. \tag{9}$$

In our solution, the VRNN generation model is treated as the $G$ function, and the discriminator function $D$ can be any neural network architecture suitable for modelling time-series data. In our work, we use a bidirectional RNN, which is the design choice of the discriminator networks in TimeGAN Yoon et al. (2019) and C-RNN-GAN Mogren (2016). Since the VRNN has an a generation model and an inference model that are both capable of producing time-series (one as a reconstruction, the other as a generation), we expose $D$ to three inputs: the real time-series data with a "real" label, the reconstruction output from the inference model with a "fake" label, and the synthetic output from the generation model with a "fake" label.

In practice, we find that exposing the discriminator to three types of data encourages a stronger distinction between real and fake data, which in turn provides the VRNN with stronger feedback. While $D$ seeks to minimize the error between classifications, the VRNN attempts to maximize this error to "fool" $D$.

### 3.3 VRNN-GAN Loss Function.

The loss functions for VRNN-GAN comprises two parts: a loss function for $D$ and the loss function of the VRNN. The training objective for the VRNN component minimizes the following expression,

$$L_{VRNN} = - \mathbb{E}\{q_{z \leq T} \mid x_{\leq T}\} \Big[ \sum_{t=1}^{T} \big( \log(p_\theta(x_t|z_\leq, x_{<t})) \tag{10}$$

$$- D_{KL}(q_\theta(z_t \mid x_{\leq t}, z_{<t})||p_\theta(z_t|x_{<t}, z_{<t}))) \Big]. \tag{11}$$

This loss is the time-step variational lower-bound from Chung et al. (2015).

Let $\overline{x}$ be a reconstruction from the Inference Model, with $x$ as the input. Let $\hat{x}$ be a sample generated by the Generation Model. Then using the VRNN component as the generator, the loss is given by

$$L_G = L_{VRNN} - \lambda_1 log(D(\overline{x})) - \lambda_2 log(D(\hat{x}), \tag{12}$$

and the discriminator loss can be written as

$$L_D = -log(1 - D(\overline{x})) - \gamma log(1 - D(\hat{x})) - log(D(x)), \tag{13}$$

where $\lambda_1, \lambda_2, \gamma$ are hyperparameters.

## 4 Experiments

In our experiments, we aimed at answering the following two research questions:

- **RQ1:** Does VRNN-GAN generate realistic data?

- **RQ2:** Does VRNN-GAN produce better data compared to other deep neural methods?

Below, we start by describing the baseline methods against which we compared the VRNN-GAN and provide details about the data that we used for the experiments and their preprocessing. Then, we briefly discuss our implementation, outline the main results, and conclude with a discussion of our findings.

### 4.1 Baseline Methods

For fair comparison, we benchmark our approach only against three baseline methods that rely on an RNN architecture and are considered state-of-the-art in their performance, namely TimeGANYoon et al. (2019), C-RNN-GAN Mogren (2016), and VRNN Chung et al. (2015). The VRNN comparison is trivial since VRNN-GAN is an extension of the training process, while C-RNN-GAN shares architectural similarities such as the choice of a bidirectional RNN as the discriminator. TimeGAN is a state-of-the-art method that has outperformed a variety of neural generative approaches on time-series generation tasks.

### 4.2 Datasets and Preprocessing

Three datasets were used to train both the VRNN-GAN and the baseline methods:

- **Autoregressive Moving Average Model (ARMA(1,1)).** The following model is constructed and recursively sampled to produce synthetic time-series data:

$$Y_t = \phi Y_{t-1} + Z_t + \theta Z_{t-1} \tag{14}$$

$$\text{where } Z_t \sim N(0, \sigma^2) \text{ and } \sigma = 1, \theta = 1. \tag{15}$$

Specifically, 1000 datapoints were sampled from this model and divided into training and test sets.

- **ECG200.** Collected by Olszewski et al. (2001), this dataset contains electrocardiogram signals (ECG) pre-sorted into train and test sets. Each set contains 100 univariate observations with a length of 97 time steps. Observations are not known to be consecutively dependent on one another.

- **Stocks.** The stock data used in Yoon et al. (2019) on Alphabet Inc. (GOOG). This is a originally a single multivariate time-series containing 3685 observations, with 6 variables per time step.

Each dataset is also split into train-test sets; the ARMA(1,1) and stock datasets were split into 80% training and 20% test data, while the ECG200 dataset had its own designated training and test sets.

Note that although traditional machine learning solutions involve the use of a validation set to detect overfitting and assess generalization, the use of validation data in typical GAN training is not particularly useful. The discriminator function is the only component that operates on training data as input, but the Generator function is the main component of interest; assessing the performance of the discriminator on unseen data would not be particularly indicative of the generator's current sampling quality. Hence, validation sets were not constructed during our experiments.

### 4.3 Implementation Details

All models and experiments were coded in the Python programming language. VRNN-GAN and VRNN was implemented in Keras Chollet et al. (2015) and Tensorflow Abadi et al. (2015). C-RNN-GAN was re-implemented in Keras and Tensorflow, while the TimeGAN implementation was modified with a few amendments to allow for Graphics Processing Unit (GPU) usage and the loading of new datasets.[1]

### 4.4 Training and Hyperparameter Tuning

The VRNN-GAN was trained using an ADAM Optimizer with a learning rate of 0.0001. For the Stock dataset, the VRNN component is initiated with 128 hidden units in the RNN, 64 in the probabilistic latent space, and 32 in the discriminator. The batch size was 64, while $\lambda_1$, $\lambda_2$ and $\gamma$ were all set to 1. For the ECG and ARMA(1,1) datasets, the Adam Optimizer was set to a learning rate of 0.001, the VRNN component was initiated with 64 hidden units in the RNN, 8 in the probabilistic latent space, and 16 in the discriminator. The batch sizes were 32, while $\lambda_1$ and $\lambda_2$ were 0 and 2, respectively, and $\gamma$ was set to 1.

### 4.5 Results

In this part, we provide our main results using standard time-series generation evaluation metrics, namely t-SNE plots for qualitative comparison, and post-hoc discriminative and predictive scores for quantitative comparison.

#### 4.5.1 t-SNE Plots

For each of the original holdout sets, the corresponding synthetic dataset from each of the models are concatenated and used to generate a t-SNE plot. A generated dataset that effectively reflects the features of the original dataset is expected to be similar in proximity in the high-dimensional space Yoon et al. (2019). Specifically, the t-SNE representations of the original and generated datasets are expected to overlap and cover the same areas in the plots if the synthetic dataset is a good reflection of the original dataset.

In Figure 3 we observe that the data generated by VRNN-GAN and TimeGAN clearly show a much stronger overlap with the ground-truth ARMA holdout set. Both methods generate synthetic datapoints in the t-SNE space that appear to follow the clusters formed by the ground-truth datapoints. The data generated by VRNN and C-RNN-GAN have little overlap with the ground-truth holdout in the t-SNE space and appear to cluster in separate spaces.

Moving on to Figure 4, we see that all methods appear to struggle to consistently overlap with the ground-truth ECG clusters. The t-SNE plots from VRNN and C-RNN-GAN appear to be centered around a single

---

[1]Code is not provided due to double-blind review process and will be available upon acceptance.

**ARMA(1,1)**

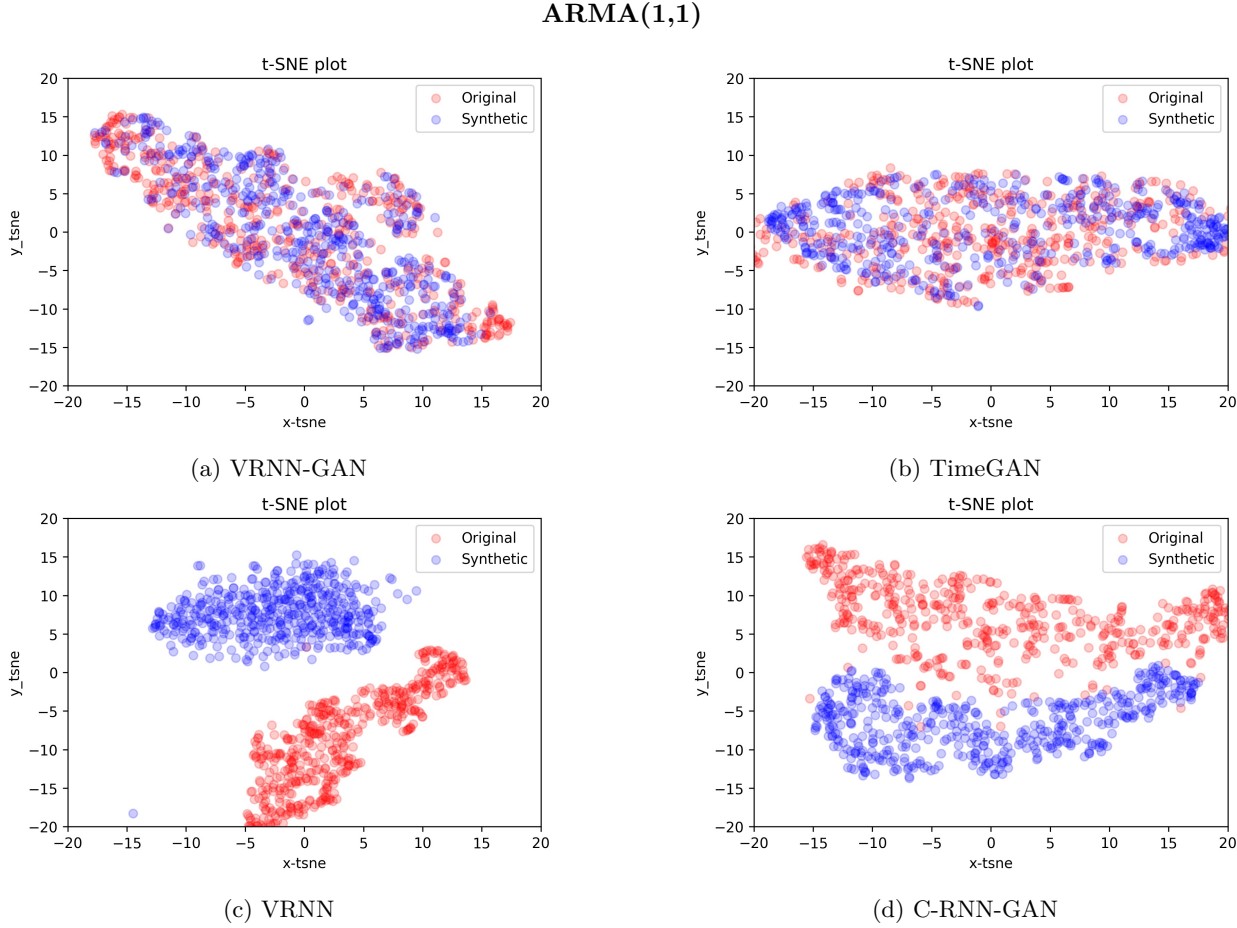

Figure 3: ARMA t-SNE comparisons

area with a slight overlap with a ground-truth cluster, yet fails to follow ground-truth clusters in other areas of the plot. t-SNE plots from the TimeGAN and VRNN-GAN generated data were able to produce more diverse points that cluster in different areas of the plot. However, they still fail to consistently follow the patterns shown in the ground-truth t-SNE points. Having said that, note that a portion of the t-SNE projections of the VRNN-GAN synthetic data were able to follow some of the clusters formed by the ground-truth points, while the majority of the TimeGAN plots are not overlapping the t-SNE points of the ground-truth data.

In Figure 5, all methods displayed some level of overlap between the synthetic and ground-truth t-SNE representations of the Stock dataset. In all four plots, the ground-truth t-SNE representations appear to group into three clusters. C-RNN-GAN appears to have the least amount of overlap, with its synthetic t-SNE representations only overlapping with one of the ground-truth clusters. t-SNE representations from the VRNN synthetic data appear to overlap with two clusters, to some degree, while a large portion of the points still group into an isolated area on the plot. TimeGAN and VRNN-GAN produce synthetic data that overlaps with all three clusters of the ground-truth data in their t-SNE representations. Yet, the t-SNE representations of the TimeGAN synthetic data form very tight patterns that cover little of the ground-truth t-SNE clusters. The synthetic data from VRNN-GAN forms more diverse clusters with larger coverage of the ground-truth data in the t-SNE representation.

**ECG**

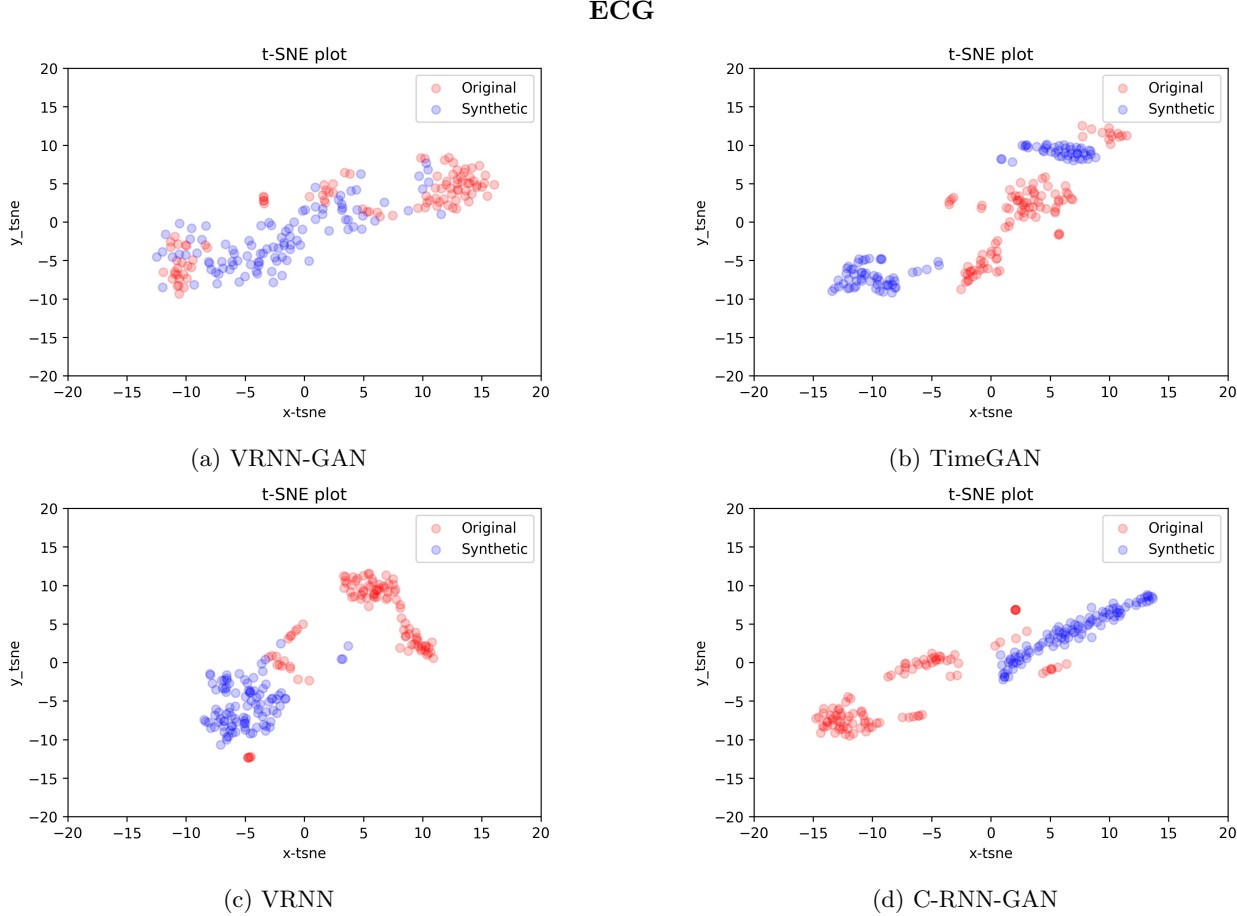

(a) VRNN-GAN                                      (b) TimeGAN

(c) VRNN                                          (d) C-RNN-GAN

Figure 4: ECG t-SNE comparisons

### 4.5.2 Post-Hoc Discriminative Score

We train a 2-layer GRU RNN model with a sigmoid output and a binary cross-entropy loss function to classify between real and generated data. The real data is given a "real" label while the synthetic data is given a "fake" label. The ground-truth holdout set is shuffled with a corresponding synthetic dataset from each of the baseline models and VRNN-GAN, and a 10-Fold cross validation approach is applied for training and evaluation. The amount of synthetic and real data is balanced. The classification error from each training-test fold pairing is averaged; the mean and standard deviations are reported.

We aim for a classification accuracy of 50% for this task, since we aim to produce synthetic data such that the performance of the binary classifier is comparable to that of a random classifier. In the event that the classifier achieves 100% accuracy, the synthetic datasets would have been unsuccessful in generating realistic data. Note that even in the latter scenario, if the classifier classifies the portions of either class correctly, the remaining portions must be incorrectly classified to achieve a total accuracy of 50%. To summarize the accuracy as a holistic score, where a lower score represents a better result, we use the Discriminative Score formulation:

$$DiscriminativeScore = |0.5 - Accuracy| \qquad (16)$$

The process and evaluation of the Discriminative Score is identical to the one proposed in Yoon et al. (2019).

According to the results in Table 1, for the ARMA(1,1) comparisons, all of the methods struggled to consistently produce synthetic data that was indistinguishable from the ground-truth holdout set. TimeGAN achieved the lowest Discriminative Score for the ARMA(1,1) data, with VRNN-GAN having the second lowest score. For the ECG dataset comparisons, VRNN-GAN reported the lowest (best) score amongst

**Stock**

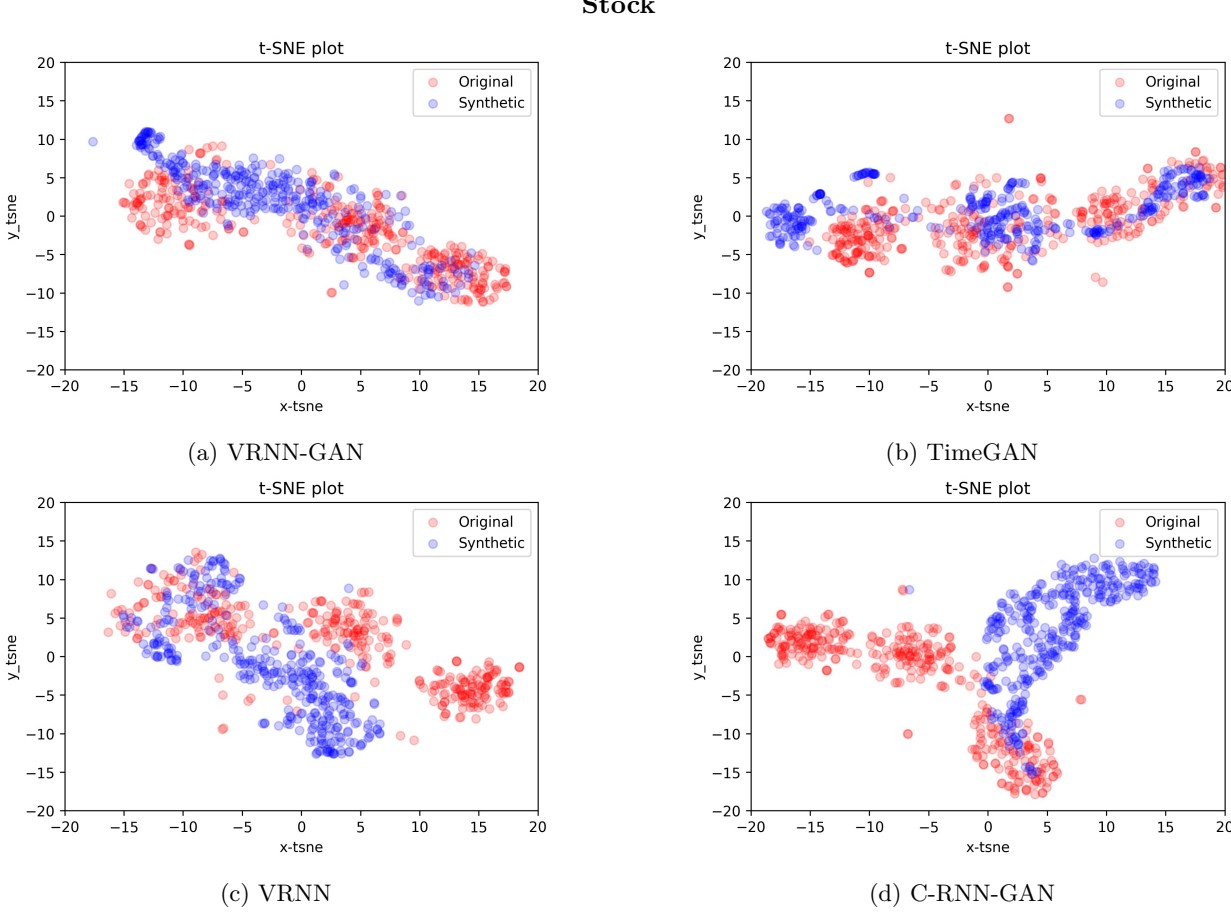

(a) VRNN-GAN

(b) TimeGAN

(c) VRNN

(d) C-RNN-GAN

Figure 5: Stock t-SNE comparisons

| Model | ARMA (1,1) | | ECG | | Stock | |
|---|---|---|---|---|---|---|
| | Mean | Std. | Mean | Std. | Mean | Std. |
| C − RNN − GAN | 0.492 | 0.006 | 0.385 | 0.105 | 0.338 | 0.026 |
| *TimeGAN* | **0.373** | 0.023 | 0.405 | 0.127 | 0.045 | 0.034 |
| *VRNN* | 0.499 | 0.003 | 0.455 | 0.042 | **0.027** | 0.022 |
| VRNN − GAN | 0.475 | 0.004 | **0.380** | 0.078 | 0.035 | 0.021 |

Table 1: Discriminative score. The lower the score, the better the performance.

all models. Lastly, for the Stock dataset comparisons, VRNN reported the lowest score and VRNN-GAN reported the second-lowest score.

### 4.5.3 Post-Hoc Predictive Score

We train a two-layer GRU RNN model for next time-step prediction using the synthetic data for training, and evaluating on the ground-truth holdout set. The last time-step of all samples in the synthetic data are isolated as prediction targets, while all previous time-steps are used as input for the GRU model. The synthetic and ground-truth holdout sets are subject to the same MinMax Scaler and re-scaling techniques used to train all baselines and VRNN-GAN. Each experiment was repeated 10 times with the same train and holdout sets for evaluation, since a standard 10-fold cross-validation is unsuitable due to the isolated nature of the training and holdout sets.

| Model | ARMA (1,1) | | ECG | | Stock | |
|---|---|---|---|---|---|---|
| | Mean | Std. | Mean | Std. | Mean | Std. |
| C − RNN − GAN | 139.95 | 25.06 | 691.24 | 23.94 | 156.21 | 1.63 |
| *TimeGAN* | 105.71 | 23.85 | 355.91 | 29.25 | 15.50 | 0.46 |
| *VRNN* | 3470.64 | 6.30 | 496.71 | 54.27 | 44.05 | 5.18 |
| VRNN − GAN | **44.41** | 14.21 | **336.54** | 64.23 | **14.26** | 0.58 |

Table 2: Predictive MAPE Score. The lower the score, the better the performance.

While Yoon et al. (2019) also use a post-hoc predictive score from an RNN predictive model for the TimeGAN evaluations, we report the Mean Absolute Percentage Error score on the re-scaled outputs from the GRU model and the unscaled ground-truth targets. This is in contrast to the Mean Absolute Error reported on the normalized outputs as done in the TimeGAN experiments. Constructing a normalized metric on the original scale of the data enables the most accurate assessment of the synthetic data in a prediction task, as it provides a fair comparison amongst the datasets.

Table 2 summarizes the results for predictive scoring. For the ARMA(1,1) results, the synthetic data generated by VRNN-GAN achieved the lowest MAPE score, beating the second lowest score by a margin of 61%. In the ECG results, the synthetic data generated by all models struggled to achieve an accurate prediction, with all methods making predictions that are on average 300% greater or less than the ground-truth. However, the synthetic data generated by VRNN-GAN still achieved the lowest score. For Stock data VRNN-GAN also achieved the best MAPE score.

### 4.6 Discussion

We discuss the results of the empirical evaluations in the context of RQ1 and RQ2.

#### 4.6.1 RQ1: Does VRNN-GAN generate realistic synthetic data?

The criteria for determining whether or not synthetic data is realistic is subjective. Brophy et al. (2021) aggregate and identify a large variety of metrics that have been applied to synthetic time-series data generated from GANs, coming to the conclusion that there is little agreement on a common set of metrics for evaluating synthetic time-series data. This is partly due to the applied nature of the task; some time-series GAN methods are designed for domain specific data, such as medical data Esteban et al. (2017) and finance Zhou et al. (2018). Domain specific metrics may be better suited and more relevant to domain specific data. As a result, the lack of agreement and large variety of evaluation methods for synthetic time-series data suggest that the meaning of "quality" in synthetic data can vary by application.

The experiments were designed to assess the validity and usefulness of generated synthetic data in the context of time-series modelling objectives, similarly to the set of experiments in Yoon et al. (2019). While each of these experiments measured a different quality measure, we believe that jointly they are relevant in evaluating the realism of synthetic data under various perspectives. The t-SNE plots and the Predictive Score measures usefulness of synthetic data in modelling tasks against the original data in clustering and prediction, respectively, while the Discriminative Score indirectly assesses the statistical and temporal dynamics of the generated data.

VRNN-GAN demonstrated good performance in terms of measures that assess the usefulness of synthetic data. Specifically the Predictive Score evaluations showed that the data generated by VRNN-GAN was useful in a very common time-series forecasting task, and indirectly shows that the training (synthetic) and testing (real) datasets are of similar distributions. For the t-SNE plots, the results were at least of the same level as the t-SNE plots generated from TimeGAN.

In the Discriminative Score evaluations, VRNN-GAN did not achieve top scores across all three datasets. However, it performed fairly well with the top score in the ECG evaluations, and the second best score in the ARMA(1,1) and the Stock evaluations. In particular, VRNN-GAN achieved its best performance on the

Stock dataset, with a score of 0.035. Although the Stock dataset is small by deep learning standards it is the most complex one compared to the other two datasets with 6 variables across 3685 observations.

Overall, we demonstrate that VRNN-GAN performs very well in tasks that measure its usefulness as substitute for the original data, and relatively well in evaluations that measure the similarity in the generated features against the original.

### 4.6.2 RQ2: Does VRNN-GAN produce better data compared to other deep generative methods?

Out of the three baseline methods, the TimeGAN model is the most sophisticated one, with unsupervised and supervised learning objectives as well as adversarial training. It contains architectures for encoding and discriminating embedding spaces as well as next-step prediction, which gives it a specific modelling advantage in the Prediction Score evaluation, a task that explicitly evaluates on next-step prediction accuracy. Despite these features, VRNN-GAN was still able to outperform TimeGAN in Predictive Score evaluations across all datasets, as well as being competitive or better in most of the other tasks, which we hypothesize can be attributed to the addition of the variational component.

Although VRNN-GAN and TimeGAN have different architectures, both methods focus on learning latent spaces from which they generate. Moreover, both methods include adversarial training. However, TimeGAN uses multiple and complex learning objectives that focus on learning specific temporal dynamics, while our model only uses the VRNN and adversarial training that is not time series-specific, which makes it more general for additional (non time-series) applications.

Overall, VRNN-GAN and TimeGAN generally outperformed VRNN and C-RNN-GAN in all evaluations, sometimes with large margins. Furthermore, the addition of adversarial training to the VRNN generally improves the quality of the synthetic data, sometimes by a large margin. We further note that VRNN-GAN is the most frequent leader across the various metric-data pairings. Thus, we conclude that VRNN-GAN outperforms the baseline methods by producing superior synthetic data in most evaluations.

## 5 Conclusion

We presented the VRNN-GAN, a novel solution to generating realistic time-series data. The solution is based on a variational recurrent neural network with adversarial training. Specifically, the variational recurrent neural network is used as a generator network, where as the discriminator network is trained to differentiate between generated and real time-series data. The variational recurrent neural network then uses this information from the discriminator network to better generate realistic time-series data.

The results of an extensive set of experiments confirm that the VRNN-GAN is competitive against state-of-the-art methods, especially on predictive tasks showing a 27.18% reduction, on average, in MAPE over the next-leading method. Throughout the experiments, we observe that our approach is competitive or better than TimeGAN, a state-of-the-art time-series GAN-based method, although the latter incorporates multiple and more complex learning objectives suitable for time-series data.

Future work involves extending our methods to larger datasets with a higher number of features. Furthermore, examining the time-step conditional variant of VRNN-GAN would be an interesting approach to observe alternative trajectories, analyze potential interventions on the generating model that would be conditional on both time-static (age, gender) and time-dynamic (temperature, day of week) features. Such an experiment could be applied as an approach to support counterfactual analysis in face of time-series data.

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
