# OpenReview forum: "VRNN’s got a GAN: Generating Time Series using Variational Recurrent Neural Models with Adversarial Training"
_TMLR — Rejected by TMLR_

### Review · Reviewer_a9c1 · 2022-12-02

**Summary Of Contributions:**

The paper tackles the important task of time series prediction. It aims to generate data which is both correct and realistic looking. For this, it combines a VAE-type generator with a GAN discriminator. The architectures used are adapted to time-series (varieties of RNNs). The approach is compared to a few VAE and GAN methods. The proposed method is better than some of the alternatives.

**Audience:**

No

**Claims And Evidence:**

Yes

**Requested Changes:**

The paper probably needs to be rejected and resubmitted as new. The new paper should include:
* A much more extensive literature review, including papers from recent years.
* A very clear discussion of the novelty compared to the recent papers
* Comparison to the most recent methods (e.g. the papers from ICML'22, NeurIPS'22)
* Much more challenging benchmarks (see the recent benchmarks used in NeurIPS papers, this is a randomly selected one - https://openreview.net/pdf?id=AyajSjTAzmg)

**Strengths And Weaknesses:**

I am not positive about this. The main strength is probably the choice of problem which is important and not sufficiently solved.

However, this paper is very incremental by nature. Its contribution is combining a VAE time series generator (VRNN) with a GAN discriminator. Both of these components are well known (and rather old), this is mentioned by the authors. The claim is that the combination is unique - which if true would be a strength. However, a simple Google search was able to retrieve this paper: "LSTM-Based VAE-GAN for Time-Series Anomaly Detection" which is roughly the same idea. Perhaps this paper has some differences from the paper I referenced but the novelty would be very small. Overall, I think the combination is not unique, and could not find the novelty in this work.

Other weaknesses: the experiments are comparing to old methods (the newest is from 2019 - and is already quite competitive with this work). The benchmarks are also quite basic, Taking the same comparisons as recent papers in NeurIPS, ICML or ICLR would have been more convincing.

---

> ### Author Response · Authors · 2023-01-31
> **Response to Reviewer a9c1**
>
> We would like to thank the reviewer for their comments and suggestions. Please find our responses below.
>
> Comment: “…this paper is very incremental by nature. Its contribution is combining a VAE time series generator (VRNN) with a GAN discriminator. Both of these components are well known (and rather old), this is mentioned by the authors. The claim is that the combination is unique - which if true would be a strength. However, a simple Google search was able to retrieve this paper: "LSTM-Based VAE-GAN for Time-Series Anomaly Detection" which is roughly the same idea. Perhaps this paper has some differences from the paper I referenced but the novelty would be very small. Overall, I think the combination is not unique, and could not find the novelty in this work.”
>
> Response:  The VRNNGAN and the LSTM-Based VAE-GAN are two different architectures, and do not constitute the same idea.  The LSTM-Based VAE-GAN relies on a non-recurrent VAE structure for generating whole sequences at once, while the VRNNGAN conditionally samples from the latent space and generates per time step.  The VRNN itself is considered a stochastic RNN, which has an inherently different formulation from a VAE + RNN combination.  Specifically, we are combining both of these methods for the task of synthetic time-series generation, and not for other classical tasks of time-series such as forecasting, prediction, classification, or anomaly detection.  We believe that our approach of combining the VRNNGAN and LSTM-Based VAE-GAN for time-series generation is a novel contribution to the field. In our revised submission, we will clearly emphasize and showcase the unique aspects of this methodology.
>
> Comment: “the experiments are comparing to old methods (the newest is from 2019 - and is already quite competitive with this work). The benchmarks are also quite basic. Taking the same comparisons as recent papers in NeurIPS, ICML or ICLR would have been more convincing.”
>
> Response:  The baselines that we used, e.g. TimeGAN is still considered one of the SOTA methods.  In our most recent search of papers that specifically focus on the synthetic time-series generation/time series data augmentation task, we have found a newer paper “Time-series Transformer Generative Adversarial Networks” (Srinivasan and Knottenbelt, 2022) that appears to beat TimeGAN on two metrics, and we will add this paper to our comparison.  Our benchmark datasets and metric choices are in-line with other papers on synthetic time-series generation or augmentation.
>
> Response to Recommendations:
> 1. We will make an additional pass over the literature and make sure to integrate additional recent papers.
> 2. We will expand the discussion in Sections 1 and 2, specifically in places where we outline the differences between our method and existing work.
> 3. We will re-examine recent works on generating time series data from the mentioned conferences. Note that not all recent papers on the subject solve the problem that we solve (generating time series data).
> 4. The benchmark datasets and performance measures that we used in our paper are the ones used by SOA methods that we outperformed for papers that solved the same problem. However, we will add the work by Srinivasan and Knottenbelt, 2022, that tackles a similar problem.

---

### Review · Reviewer_4Px4 · 2023-01-04

**Summary Of Contributions:**

The paper propose VRNN-GAN, which a generative model for generating time-series data. The model consists of a variational recurrent
neural network as a generator network, and a discriminator network that differentiates between generated and real time-series data. Results show some improvements over previous time-series generative models.

**Audience:**

Yes

**Claims And Evidence:**

Yes

**Requested Changes:**

1. Add some motivations for designing the model

2. Draw connections between the model and some VAE-GAN hybrids in image generation


**Strengths And Weaknesses:**

Strengths:

1. The idea is straightforward and easy to follow. It is a natural extension over VRNN by adding adversarial objective to improve sample quality.

2. The training objective is clear and well presented.

3. Empirical results show improvements over models like VRNN

Weaknesses:

1. The motivation of the model design is not well explained. Section 3 goes straight into the model details without explaining why the design is a good idea.

2. In image generation, there is a line of work that combines VAE and GAN, including the one with discriminator after the VAE generator (such as https://arxiv.org/pdf/1512.09300.pdf). The model is a direct extension of the image generation model. Some connections can be made.

3. I am not very familiar with the literature of time series, but the dataset used in experiment section seems to be small scale and simple.

---

> ### Author Response · Authors · 2023-01-31
> **Response to Reviewer 4Px4**
>
> We would like to thank the reviewer for their comments and suggestions. Please find our responses below.
>
> Comment: “The motivation of the model design is not well explained. Section 3 goes straight into the model details without explaining why the design is a good idea.”
>
> Response: We will extend the motivation in introduction and the beginning of Section 3 to better motivate our design choices, including the architecture and the unique combination of VAE-RNN-GAN.
>
> Comment: “In image generation, there is a line of work that combines VAE and GAN, including the one with discriminator after the VAE generator (such as https://arxiv.org/pdf/1512.09300.pdf). The model is a direct extension of the image generation model. Some connections can be made.”
>
> Response: Indeed, we cited the paper, and we will make sure that in the next version we will emphasize the connection in a more coherent fashion.
>
> Comment: “I am not very familiar with the literature of time series, but the dataset used in experiment section seems to be small scale and simple.”
> Response: We have used a standard collection of datasets as the most recent SOA papers that we outperformed in the paper. We will emphasize this statement in the evaluation section.

---

### Review · Reviewer_HS7p · 2023-01-21

**Summary Of Contributions:**

The authors propose VRNN-GAN, a system consisting of the VAE loss, applied on an RNN, trained adversarially to generate realistic-looking synthetic time series data. This synthetic data is to be used to train models in settings where data is scarce, or where anonymity protection is key.

They evaluate their results qualitatively via t-sne plots, and quantitatively via a post-hoc discriminative and predictive score.

They demonstrate via these evaluations that their model seems to be on par or outperform the competition.

**Audience:**

Yes

**Claims And Evidence:**

Yes

**Requested Changes:**

I would request that the authors do the following changes:

Abstract:

Feels overly wordy and as a result, distracts from the narrative the authors are trying to convey.

I would recommend using a simpler, punchier formula for the abstract. Consider the one at https://www.easterbrook.ca/steve/2010/01/how-to-write-a-scientific-abstract-in-six-easy-steps/

Make sure that you are clearly explaining what current methods lack, and what your method is adding to the mix, to produce gains along an axis that is important.

Introduction:

Introduction:

You probably want to use /citep for the references here as they seem out of place as they currently are. The writing here also feels a bit clumsy, consider rephrasing for more clarity/precision.

I find that the problem itself, that is, generating synthetic time series data as a means to training better models in data-scarce settings not very convincing, but can be useful to some degree. I would recommend clearly stating work where time-series generative models have been used successfully in improving the performance of models in data-scarce situations, as well as situations where the anonymity of the source data must be protected.

Question: If a generative model must be trained on the source data that need to remain anonymous, then doesn't that defeat the idea of retaining anonymity, since the data have already been given to the generative model training process?

It would be worth making a clearer case for the benefits of generative models in time series situations as a means of augmentation. I do believe that there are many, but the introduction doesn't do a good job explaining this.

Figure 3 to 5: Why is each model applied on a separate held out set? This makes me wonder whether the results look better or worse based on the held out sets unique distribution. It would make more sense to showcase how each model performs on the **same** held out test set.


**Strengths And Weaknesses:**

Strengths:

Domain/Model: The system proposed by the authors is novel in the sense that it hasn't been used before on this particular problem. However, its constituent parts have been previously used, and VRNNs in GAN setups have also been used before.

Evaluation: The evaluation is enough to demonstrate usefulness of the method, and contains multiple different metrics.

Improvements: The method proposed seems to be on par, or better than the SOTA in this problem.

Weaknesses:

As stated earlier, the application of this particular model type on this problem is novel. However the model itself or its constituent parts is not novel.

The evaluation was enough for a basic proof of usefulness, but it's far from thorough or 'good' in the sense that more experiments are required for one to state a conclusion that the method does outperform other methods statistically significant enough that one could say that this method may retain it's ranking as the best method in a real world setting.

The writing overall feels clumsy, wordy and at times imprecise.

---

> ### Author Response · Authors · 2023-01-31
> **Response to reviewer HS7p**
>
> We would like to thank the reviewer for their comments and suggestions. Please find our responses below.
>
> Comment: “Abstract: Feels overly wordy and as a result, distracts from the narrative the authors are trying to convey. I would recommend using a simpler, punchier formula for the abstract. Consider the one at https://www.easterbrook.ca/steve/2010/01/how-to-write-a-scientific-abstract-in-six-easy-steps/
> Make sure that you are clearly explaining what current methods lack, and what your method is adding to the mix, to produce gains along an axis that is important.”
>
> Response: Thank you - we will update the abstract as the reviewer suggests.
>
> Comment: “Introduction: You probably want to use /citep for the references here as they seem out of place as they currently are. The writing here also feels a bit clumsy, consider rephrasing for more clarity/precision.”
>
> Response: We will use the \citep command to fix the references.
>
> Comment: “I find that the problem itself, that is, generating synthetic time series data as a means to training better models in data-scarce settings not very convincing, but can be useful to some degree. I would recommend clearly stating work where time-series generative models have been used successfully in improving the performance of models in data-scarce situations, as well as situations where the anonymity of the source data must be protected.”
>
> Response: The argument for generating synthetic time series data is the same for any type of motivation in deep generative models that have been developed in recent years.  Deep generative models have been applied to images, text, tabular data, etc. with the common motivation that deep learning models require large datasets for learning, and that large datasets are often not available in many scenarios.  We believe that the general argument data augmentation methods is applicable to our method
>
> Comment: “Question: If a generative model must be trained on the source data that need to remain anonymous, then doesn't that defeat the idea of retaining anonymity, since the data have already been given to the generative model training process? It would be worth making a clearer case for the benefits of generative models in time series situations as a means of augmentation. I do believe that there are many, but the introduction doesn't do a good job explaining this.”
>
> Response: Our scenario pertains to the situation where small datasets are made available but the process of procuring such a dataset is difficult for reasons related to anonymity and general privacy processes.  Such a situation would benefit from a generative model that could produce training samples from the same distribution.  We will aim to better clarify this point in the introduction section.
>
> Comment: “Figure 3 to 5: Why is each model applied on a separate held out set? This makes me wonder whether the results look better or worse based on the held out sets' unique distribution. It would make more sense to showcase how each model performs on the same held out test set.”
>
> Response: Each model is applied to the same hold out set across the different datasets.  However, the orientation of the plots may depend on the results of the test method.  We will ensure to state this so to clarify the issue.

---

### Review · Reviewer_Kqr8 · 2023-01-27

**Summary Of Contributions:**

The paper focuses on the problem of generating realistic time series. It proposes a GAN based framework where VRNN is used as an encoder and Bi-directional RNN as decoder. Experiments show that the proposed approach can perform better in specific cases.

**Audience:**

No

**Claims And Evidence:**

No

**Requested Changes:**

I would like to see more experiments to evaluate the usefulness of the approach. It would be useful to show if synthetic data can improve clinical research where datasets are generally small, or any downstream use case as an evidence.

**Strengths And Weaknesses:**

### Strengths
- The paper focuses on an important problem of time series generation which can be useful in several practical use cases.
- The paper is easy to follow.


### Weaknesses
- The proposed approach is lacking novelty as this is a straight-forward application of two well-known approaches.
- Experiments are not compelling and based on the results, I'm not convinced that this approach performs better than baseline methods.
- The paper motivates in the introduction that datasets in the clinical research are smaller because of privacy and other reasons. Can this approach mitigate this issue? The paper should show how the generated time series could be useful for a downstream task.
- Most of the details in Section 3 are unnecessary, for example description of VRNN and adversarial training is not required. The paper should focus on the main contributions and discuss that in detail.

---

> ### Author Response · Authors · 2023-01-31
> **Response to reviewer Kqr8**
>
> We would like to thank the reviewer for their comments and suggestions. Please find our responses below.
>
> Comment: The proposed approach is lacking novelty as this is a straight-forward application of two well-known approaches. Experiments are not compelling and based on the results, I'm not convinced that this approach performs better than baseline methods. The paper motivates in the introduction that datasets in the clinical research are smaller because of privacy and other reasons. Can this approach mitigate this issue? The paper should show how the generated time series could be useful for a downstream task.
>
> Response: We will provide a case study analysis with a real dataset from a clinical setting to show the effectiveness of this method. The dataset comes from a cancer hospital in the United States. We recently received permission to use the data in future publications. Hopefully, this will clarify the doubts surrounding the motivation for the problem and applicability of the approach.
>
> Comment: Most of the details in Section 3 are unnecessary, for example description of VRNN and adversarial training is not required. The paper should focus on the main contributions and discuss that in detail.
>
> Response: We will add more comments relating to the motivation behind our approach as it pertains to VAE-GAN hybrids and emphasize the main contributions more in this section. We believe that adding these details will show that the VRNNGAN as we designed it is indeed different (a message that did not come across in the current version) and would justify the detailed description in Section 3.

---

### Decision · Action_Editors · 2023-03-09

**Recommendation:** Reject

**Comment:**

4 reviewers reviews the paper. The paper has 3 major issues:

1) The experimental investigation is very limited and not convincing. More recent methods as well as datasets should be tested to make convincing results. In particular, Yoon et al. 2019 (TimeGAN) tests on 4 time-series datasets, whereas this manuscript only selected one of those dataset for testing. Rev a9c1 also recommends some other datasets used in recent papers. As for methods, there are recent approaches for time-series generation, e.g., [GT-GAN: General Purpose Time Series Synthesis with Generative Adversarial Networks, NeurIPS'22], ["Time-series Transformer Generative Adversarial Networks", Srinivasan and Knottenbelt, 2022]

2) A major use case of the proposed method is to generate training data in data-scarce situations. However, this is not tested at all. A case study would be helpful to convincingly show that the method works in practice.

3) The presentation needs significant improvement.  The differences with previous methods needs to be better explained (e.g., LSTM-based VAE GAN, image VAE GANs), and more recent works included.

No updated manuscript was submitted for the author response, so it is unclear how the proposed changes to the paper will look in the end. Overall the manuscript needs significant revisions.


**Audience:**

All reviewers think that the topic is interesting and could be useful in practice. However, the experiments are too limited (too few datasets, and too few comparison methods), to make a definitive conclusion about the merits and demerits of the proposed method as compared to alternatives. Thus, the current findings in the paper will be of limited interest.

**Claims And Evidence:**

Several claims are not supported well enough.
1) "We show that VRNN-GAN achieves the best predictive score across all methods and yields competitive results in other well-established performance measures compared to the state-of-the-art. -- Reviewers thought that are too few methods and datasets in the experiments for this claim to be supported. (Rev HS7p, Kqr8, a9c1)

2) motivation of preserving privacy by using a synthetic dataset rather than sensitive data - Rev HS7p noted that the data needs to be used to train the synthetic data generator, so it defeats the purpose of retaining anonymity. Thus the proposed method cannot use this motivation of preserving privacy.

3) motivation of using generated time series in practice in data-scarce situations - there were no experiments to support this. (Rev Kqr8)